# Genotype and Phenotype Analyses of a Novel *WFS1* Variant (c.2512C>T p.(Pro838Ser)) Associated with DFNA6/14/38

**DOI:** 10.3390/genes14020457

**Published:** 2023-02-10

**Authors:** Hedwig M. Velde, Xanne J. J. Huizenga, Helger G. Yntema, Lonneke Haer-Wigman, Andy J. Beynon, Jaap Oostrik, Sjoert A. H. Pegge, Hannie Kremer, Cris P. Lanting, Ronald J. E. Pennings

**Affiliations:** 1Department of Otorhinolaryngology, Radboudumc, 6525 GA Nijmegen, The Netherlands; 2Donders Institute for Brain, Cognition and Behaviour, Radboudumc, 6525 GA Nijmegen, The Netherlands; 3Department of Human Genetics, Radboudumc, 6525 GA Nijmegen, The Netherlands; 4The Radboud Institute for Molecular Life Sciences, Radboud University Medical Center, 6525 GA Nijmegen, The Netherlands; 5Department of Medical Imaging, Radboudumc, 6525 GA Nijmegen, The Netherlands

**Keywords:** hereditary hearing loss, *WFS1*, DFNA6/14/38, human genetics, autosomal dominant hearing loss, low-frequency sensorineural hearing loss, genotype, phenotype, likely pathogenic variant

## Abstract

The aim of this study is to contribute to a better description of the genotypic and phenotypic spectrum of DFNA6/14/38 and aid in counseling future patients identified with this variant. Therefore, we describe the genotype and phenotype in a large Dutch–German family (W21-1472) with autosomal dominant non-syndromic, low-frequency sensorineural hearing loss (LFSNHL). Exome sequencing and targeted analysis of a hearing impairment gene panel were used to genetically screen the proband. Co-segregation of the identified variant with hearing loss was assessed by Sanger sequencing. The phenotypic evaluation consisted of anamnesis, clinical questionnaires, physical examination and examination of audiovestibular function. A novel likely pathogenic *WFS1* variant (NM_006005.3:c.2512C>T p.(Pro838Ser)) was identified in the proband and found to co-segregate with LFSNHL, characteristic of DFNA6/14/38, in this family. The self-reported age of onset of hearing loss (HL) ranged from congenital to 50 years of age. In the young subjects, HL was demonstrated in early childhood. At all ages, an LFSNHL (0.25–2 kHz) of about 50–60 decibel hearing level (dB HL) was observed. HL in the higher frequencies showed inter-individual variability. The dizziness handicap inventory (DHI) was completed by eight affected subjects and indicated a moderate handicap in two of them (aged 77 and 70). Vestibular examinations (*n* = 4) showed abnormalities, particularly in otolith function. In conclusion, we identified a novel *WFS1* variant that co-segregates with DFNA6/14/38 in this family. We found indications of mild vestibular dysfunction, although it is uncertain whether this is related to the identified *WFS1* variant or is an incidental finding. We would like to emphasize that conventional neonatal hearing screening programs are not sensitive to HL in DFNA6/14/38 patients, because high-frequency hearing thresholds are initially preserved. Therefore, we suggest screening newborns in DFNA6/14/38 families with more frequency-specific methods.

## 1. Introduction

Hereditary non-syndromic hearing loss (NSHL) is a genetically and clinically highly heterogeneous disorder. To date, causative variants have been reported in 124 genes [1]. The most common modes of inheritance are autosomal recessive and autosomal dominant, occurring in about 80% and 20% of cases of early-onset NSHL, respectively [2]. Rarer modes of inheritance are X- or Y-linked and mitochondrial. NSHL is classified according to the inheritance pattern, with autosomal recessively inherited NSHL notated as DFNB and dominantly inherited NSHL as DFNA. This is followed by a subtype number that indicates the order in which the subtypes and associated chromosomal loci were first described.

The DFNA6, DFNA14 and DFNA38 loci were initially reported separately but were later found to be associated with pathogenic variants in the same gene: *WFS1* [3,4,5,6]. Over 40 variants within this gene are known to cause LFSNHL [7]. In addition to *WFS1*, *DIAPH1* (OMIM 602121, DFNA1) and *CCDC50* (OMIM 611051, DFNA44) are associated with LFSNHL [8]. DFNA6/14/38, however, is the most common form of hereditary LFSNHL [9]. Pathogenic *WFS1* variants cause a spectrum of different disorders, including Wolfram syndrome (OMIM 222300), Wolfram-like syndrome (OMIM 614296) and DFNA6/14/38 (OMIM 600965). The autosomal recessively inherited Wolfram syndrome is characterized by childhood-onset diabetes mellitus, optic atrophy, high-frequency sensorineural hearing loss (SNHL), diabetes insipidus and neurological and psychiatric symptoms [10,11,12,13]. It is a rare condition with a prevalence in the United Kingdom of 1 in 770,000 [14]. Wolfram-like syndrome presents with similar symptoms of diabetes mellitus, optic atrophy and SNHL, but is inherited in an autosomal dominant manner [15]. Finally, DFNA6/14/38 is a non-syndromic form of autosomal dominantly inherited SNHL and, in contrast to the phenotype of Wolfram and Wolfram-like syndrome, mainly affects the lower frequencies [5,6]. Other *WFS1*-associated disorders are isolated forms or combinations of the previously mentioned symptoms, such as autosomal dominantly or recessively inherited non-syndromic diabetes mellitus [16,17], autosomal recessively inherited syndromic (with diabetes and SNHL) or non-syndromic cataract [18,19], and autosomal recessively inherited optic neuropathy and SNHL [20].

All variants indisputably associated with autosomal dominantly inherited disease, both syndromic (Wolfram-like syndrome) and non-syndromic (DFNA6/14/38), are missense variants or small in-frame deletions located in exon 8. This is the last and largest exon of *WFS1* and codes for the transmembrane domains and the C-terminal domain of the protein (UniProt accession no. O76024) [21,22]. Although missense variants in other exons have been reported, genetic analyses or reported data were limited, and the reported variants are currently classified as benign, likely benign or of unknown significance in the Leiden Open Variation Database 3.0 and ClinVar [23,24,25,26,27,28]. Heterozygous carriers of truncating variants generally do not have HL; a heterozygous inactivating variant thus does not lead to a phenotype. This suggests that variants associated with Wolfram-like syndrome or DFNA6/14/38 are non-inactivating variants with a dominant-negative effect in which the mutant protein impairs the function of the wild-type protein [29]. Alternatively, the aberrant protein has a toxic gain-of-function effect. In contrast, the autosomal recessively inherited Wolfram syndrome results from a loss-of-function mechanism due to inactivating variants in both alleles leading to no or strongly reduced (functional) protein production [10,29].

It is not yet understood why some variants lead to syndromic and others to non-syndromic disease or why DFNA6/14/38 is characterized by LFSNHL, while in most syndromic forms, the higher frequencies are typically affected. Insights into correlations between genotypes and phenotypes are essential for genetic counseling of patients and their families. In addition, they are fundamental to possible future development and evaluation of genetic therapies. In this report, we describe a novel *WFS1* variant (Chr4(GRCh37):g.6304034C>T NM_006005.3:c.2512C>T NP_005996.2:p.(Pro838Ser)) that co-segregates with LFSNHL in a large Dutch–German family.

## 2. Materials and Methods

This study was approved by the Medical Research Ethics Committee East Netherlands (registration number: NL33648.091.10) and conducted in accordance with the Declaration of Helsinki. Written informed consent was received from all individuals or their legal representatives.

### 2.1. Genetic Analyses

The proband of family W21-1472 (subject IV:07) presented with LFSNHL at our clinic for genetic testing of HL. Analysis of a gene panel for hearing impairment (version DG 2.18, containing 206 genes) was performed on genetic data generated with exome sequencing (Agilent SureSelectQXT Human All Exon v5 enrichment Kit) in the ISO15189 accredited Genome Diagnostic Laboratory of the Radboud university medical center (Nijmegen, the Netherlands) according to routine diagnostic procedures [30]. Variants in genes from this panel with a frequency below 5% in the dbSNP database and below 1% in an in-house database (consisting of exome sequencing data of ~25,000 individuals, mainly of Dutch origin) and within the exon or the intronic position of −8 to +3 were interpreted and classified following existing variant classification guidelines established by the Association for Clinical Genetic Science and the Dutch Society of Clinical Genetic Laboratory Specialists [31]. After identification of a novel *WFS1* variant, this family study was initiated.

The following in silico prediction tools were used to further assess the potential pathogenicity of the variant: CADD [32], REVEL [33] and SpliceAI (threshold: ≥0.1/1) [34]. The American College of Medical Genetics and Genomics (ACMG) Standards and Guidelines and its specifications for HL [35,36] were used to classify the variant. Testing for co-segregation of the variant with hearing loss was performed in 18 relatives by PCR and subsequent Sanger sequencing according to standard protocols.

### 2.2. Clinical Evaluation

To comprehensively describe the subjects’ phenotypes, we collected general demographic and medical data, including age, descent, comorbidities, medication use and medical history. Specifically, age of onset, type and progression of audiological and vestibular symptoms were assessed. The medical history of the ear focused on recurrent ear infections, previous noise exposure, use of ototoxic medication, meningitis, severe head injuries and ear surgeries. In addition to general and audiovestibular clinical data, symptoms of optic atrophy, diabetes mellitus, diabetes insipidus and psychiatric disorders were evaluated. These data were collected through anamnesis and a clinical questionnaire on general audiovestibular symptoms. The dizziness handicap inventory (DHI), consisting of an emotional (E, maximum score of 36), functional (F, maximum score of 36) and physical (P, maximum score of 28) subdomain, was used to quantify possible vestibular symptoms [37,38]. For children between 5 and 12 years, the DHI for patient caregivers (DHI-PC) was deployed [39]. Otoscopic assessment, pure tone and speech audiometry and clinical vestibular examination (Head Impulse Test) were performed in all subjects. Results of previous audiometric examinations were retrieved from other centers when available. Finally, MR imaging of the inner ear was performed in the proband to assess potential anatomical anomalies.

Pure tone audiometry was performed in a soundproof cabin. The pure tone audiometry thresholds were determined at 0.25 to 8 kHz, according to current standards. Subjects were considered affected when thresholds higher (worse hearing) than the age- and gender-specific 95th percentile (International Organization for Standardization, ISO 7029:2017 [40]) were observed in pure tone audiometry for at least three frequencies of the best hearing ear. HL was considered asymmetric if a difference of at least 10 decibel hearing level (dB HL) was observed between both ears at two or more frequencies [41]. A mean audiogram with 95% confidence interval was established. We performed cross-sectional linear regression analyses on threshold-by-age data from the patients’ most recent visits were used to evaluate the progression of HL in this family. Based on this, we obtained age-related typical audiograms (ARTA) according to previously described methods [42]. In evaluation of the progression, we also corrected for presbycusis, i.e., for age- and gender-specific median norms (50th percentile, ISO 7029:2017 [40]).

Speech audiometry was performed with a phonetically balanced standard Dutch consonant-vowel-consonant word list [43]. It was only conducted on native Dutch subjects and German subjects with sufficient Dutch speech. Speech recognition thresholds (SRTs) in speech audiometry were defined as the level at which a score of 50% correct was achieved at the monosyllable test [44]. Maximum phoneme recognition scores (speech recognition scores, SRS) were derived from performance intensity plots of the individual speech audiogram. Digits in noise (DIN) tests were performed as described previously; SRTs in DIN tests were defined as the signal-to-noise ratio at which 50% of the presented triplets were correctly reproduced [45]. Audiometry scores were reported as mean and standard error of mean (±SEM) for normally distributed data or as median and the interquartile range (IQR) for non-normally distributed data. Calculations and analyses of audiometric data were performed with RStudio (version 1.4.1106).

To assess vestibular function in this family, we performed vestibular examination in four subjects of different ages, both with and without reported vestibular symptoms. It was not logistically feasible to achieve this in all subjects. The oculomotor function was assessed by oto-neurological motor tests (smooth pursuit, gaze, optokinetic nystagmus, fixation suppression, and saccade tests). The function of both horizontal semicircular canals was assessed by caloric irrigation and rotational chair tests (bi-directional velocity step). Video Head Impulse Tests (vHIT) were used to determine the function of all semicircular canals. Finally, ocular and cervical Vestibular Evoked Myogenic Potential (o-VEMP and c-VEMP) were used to evaluate bilateral otolith function. All tests were executed according to the methodology described previously [46].

## 3. Results

The proband of family W21-1472 (subject IV:07) presented at our clinic with bilateral, symmetric, moderate LFSNHL since childhood. The family pedigree (Figure 1) suggested a dominant inheritance pattern of HL. Nineteen subjects, including the proband, participated in this study.

### 3.1. Genetic Analyses

Exome sequencing identified a novel *WFS1* missense variant (Chr4(GRCh37): g.6304034C>T NM_006005.3:c.2512C>T NP_005996.2:p.(Pro838Ser)) in heterozygous state in the proband (IV:07). No clinically relevant pathogenic variants, likely pathogenic variants or variants of unknown significance were identified in other genes associated with HL. The variant is not present in the gnomAD database (v3.1.2) and was initially classified as a variant of unknown significance. It was predicted to be pathogenic by both used in silico tools (CADD 27.3; REVEL 0.926). There was no effect on splicing predicted by SpliceAI. After confirmation of co-segregation with hearing loss, the variant was reclassified as “Likely pathogenic” according to the ACMG Standards and Guidelines and its specifications for HL (PM2, PP1_strong, PP3, PP4).

The presence of the c.2512C>T p.(Pro838Ser) variant was addressed in 17 relatives of the proband (Figure 1). All 10 tested relatives with LFSNHL (III:02, III:04, III:11, III:13, III:25, III:27, IV:03, IV:09, IV:13, V:04; Figure 1 and Figure 2) were found to be heterozygous for the variant. Genetic testing was rejected by the parents in one young subject (V:01) with LFSHNL. All relatives who tested negative for the variant had normal hearing (III:08, III:17, III:18, III:22, IV:05, IV:12) or SNHL atypical for DFNA6/14/38 (III:21, Appendix A).

### 3.2. Clinical Evaluation

The age of the participating subjects ranged from 5 to 77 years (Appendix A). Thirteen subjects reported HL, 11 of whom used hearing aids. Six had risk factors for acquired HL, e.g., recurrent ear infections or previous noise exposure. The self-reported onset of HL ranged from birth to 50 years of age, but was mainly in the first decade of life. In the two youngest subjects (V:01 and V:04) neonatal hearing screening was performed, and both passed this screening. Seven subjects reported tinnitus as an additional symptom to their HL. Three affected subjects (III:02 aged 77, III:04 aged 72 and III:11 aged 70) reported vestibular symptoms on the general questionnaire. The DHI was completed by seven of 10 affected adults (III:02, III:11, III:13, III:25, IV:07, IV:09, IV:13) and by five out of seven unaffected adults (III:08, III:17, III:18, III:22, IV:05). The median score of affected adults was 14 (IQR: 34) and that of unaffected adults (in this cohort) 0 (IQR: 4). The difference in DHI scores between affected and unaffected adults was not statistically significant (Independent Samples *t* Test, *p* = 0.106). The DHI-PC was completed for one affected child (V:04); she scored 0. The scores of subjects III:02 and III:11 were both classified as a moderate handicap (subdomain scores: III:02 E 14/36, F 16/36, P 8/28; III:11 E 2/36, F 10/36, P 22/28). All other scores were classified as no handicap. There were no reported indications of optic atrophy, diabetes mellitus or psychiatric disorders in this family. One subject (V:01) had previously been diagnosed with familial diabetes insipidus; in contrast to the HL, this affected the maternal side of her family.

In most cases, the physical examination of the ears was normal; (mild) myringosclerosis was observed in five subjects. No abnormalities were observed in the Head Impulse Test.

Six participants had normal hearing (III:08, III:17, III:18, III:22, IV:05, IV:12) on audiometry. Thirteen subjects had bilateral, symmetrical SNHL, 12 of whom had LFSNHL typical for DFNA6/14/38 (Figure 2A). All HL was classified as purely sensorineural. Pure tone averages (PTAs) at 0.5 to 2 kHz (PTA_0.5–2 kHz_) of the 12 DFNA6/14/38 subjects ranged from 35 to 74 dB HL (mean 55.6 ± 3) (Appendix A). The HL was asymmetric in subject III:04. Since the degree of asymmetry was minimal, all other subjects had symmetric HL and based on previous studies, there are no reasons to assume that pathogenic *WFS1* variants cause asymmetric loss, we averaged the thresholds of both ears for further statistical analyses. The mean audiogram showed average hearing thresholds in the low frequencies (0.25–2 kHz) of about 50–60 dB HL and in the high frequencies (4–8 kHz) of about 30–35 dB HL (Figure 2B). The variation in hearing thresholds is larger in the high frequencies than in the low frequencies. The ARTA display progression of HL in the high frequencies (Figure 2C). However, linear regression analysis demonstrated no statistically significant progression for any frequency, neither with nor without correction for presbycusis.

Speech audiometry and DIN tests were performed on 12 and seven affected subjects, respectively. The mean maximum phoneme recognition scores of both ears were 90% or higher in seven out of 12 subjects (median 97 (IQR: 15)), despite the elevated mean SRTs of 46 (±4) dB HL. Regression analysis of the SRT as function of three different PTAs showed that the SRT in this family could be best predicted by the PTA of 1, 2 and 4 kHz (R^2^ = 0.63, compared to R^2^ = 0.61 for PTA_0.5–4 kHz_ and R^2^ = 0.58 for PTA_0.5–2 kHz_) (Appendix A).

Cross-sectional analyses of performance expressed as a maximum phoneme recognition score of 90% relative to age and PTA_0.5–2 kHz_ showed an onset age of deterioration of speech perception of 58 years and an onset level of 58 dB HL (Figure 3). In DIN tests, the SRTs for all subjects were worse than the norm (mean −3.2 ± 1 dB HL) [45].

Four subjects were selected for vestibular examination, two elderly subjects with reported vestibular symptoms (III:04 and III:11) and two middle-aged subjects without reported vestibular symptoms (IV:07 and IV:09) (Table 1). The results of the caloric irrigation test and the vHIT were within the normal range in all subjects, indicating normal vestibulo-ocular reflex function of all semicircular canals. Mild hyperreactivity on the rotary chair test was observed in subjects III:11 and IV:07. Outcomes of VEMP testing suggest an otolith dysfunction in subjects III:04, III:11 and IV:09.

No inner ear abnormalities were demonstrated by MR imaging in the proband, including no cochlear or vestibular hydrops.

## 4. Discussion

This study describes the identification of a novel likely pathogenic *WFS1* variant (c.2512C>T p.(Pro838Ser)) associated with DFNA6/14/38 in 12 subjects with LFSNHL. The lower frequencies (0.25–2 kHz) are primarily affected with average thresholds of about 50 to 60 dB HL. Mean hearing thresholds in the higher frequencies (4–8 kHz) are about 35 dB HL. There was no statistically significant progression in any frequency. Although the self-reported age of onset varied from congenital to 50 years, LFSNHL was present in both young subjects (V:01 aged 5, V:04 aged 11) in whom the variant was identified. These two young individuals passed neonatal hearing screening. Speech recognition is relatively mildly affected compared with PTA scores. DHI scores indicated a moderate handicap in two out of seven affected adults who completed this questionnaire. Vestibular test results mainly showed abnormalities in otolith function on VEMP testing, although performed in only four subjects.

The wide variability in self-reported onset age of HL in DFNA6/14/38 in this study is reflected in a similar range from congenital to over 40 years reported in literature [47,48]. Retrospective estimation of the age of onset of HL is difficult and in DFNA6/14/38, the relatively mild presentation and the possibility of compensation through a relatively preserved high-frequency perception may lead to an overestimation, i.e., a retrospectively estimated later age of onset than the actual age of onset of HL [8,49]. It is important to note that nearly all previously described children with DFNA6/14/38 have an early age of onset and LFSNHL on audiometry. Only one study reported a *WFS1* variant identified in an eight-year-old without apparent HL [50]. For this case, it is remarkable that the reported bilateral thresholds are similar at all frequencies. While this may be true and coincidental, practice hardly shows exactly equal thresholds, raising questions about the reliability of the audiogram obtained in this young patient. Based on our observations in the youngest subjects, in whom LFSNHL was diagnosed as early as age 5 (V:01) and 6 (V:04), we assume that the onset of HL in DFNA6/14/38 is early, probably congenital. Since the method of the Dutch neonatal hearing screening is based on click-evoked otoacoustic emissions (OAEs), which are most sensitive at 1 to 4 kHz, subjects with this phenotype will not be easily identified because of preserved hearing in the higher frequencies. This may thus lead to a ‘pass’ criterion, in this case a false-negative result for HL in the low frequencies. Therefore, we propose screening newborns in DFNA6/14/38 families with a frequency-specific test, such as the narrow-band chirp Auditory Brainstem Response (ABR) test or the Auditory Steady-State Response (ASSR) test. This is in line with an earlier recommendation by Lesperance et al. [47]. Alternatively, newborns in DFNA6/14/38 could be genetically tested.

Although there is inter-familial variability, DFNA6/14/38 ARTA are highly recognizable and the ARTA of this family are no exception [51,52]. The observed average hearing thresholds and greater variability in the high frequencies than in the low frequencies are also consistent with previous findings. The most recent European/American DFNA6/14/38 audio profile (2020), based on 337 audiograms from 57 families, shows hearing thresholds of approximately 30 to 70 dB HL in the low (0.25–2 kHz) and 20 to 80 dB HL in the high (4–8 kHz) frequencies [53]. We cannot fully explain the variability of hearing thresholds in either low or high frequencies in this family by the available clinical data. Age does not seem to be an important factor; after correction for presbycusis, the hearing threshold ranges were between 28 and 66 dB HL (PTA_0.25–2 kHz_) and between −6 and 38 dB HL (PTA_4–8 kHz_). Thus, in addition to inter-familial phenotypic variability in DFNA6/14/38, we observed intra-familial variation. This has also been reported in other forms of autosomal dominantly inherited HL [54,55].

The onset level of deterioration of speech perception is comparable to what was previously demonstrated in other DFNA6/14/38 families; in contrast, the onset age of deterioration of speech perception is considerably higher [51]. The relatively good speech perception in DFNA6/14/38 patients may be attributed to the preservation of hearing in the higher frequencies and its importance for speech and language development [56]. Consistent with this, we found that the observed SRTs are best described by the higher frequency PTAs (1–4 kHz); DFNA6/14/38 patients are remarkably mildly affected in terms of their speech recognition when the HL is limited to the low frequencies. This is in contrast to presbycusis (mildly down sloping audiogram configuration) in which the SRTs are best predicted by lower frequency PTAs (0.5–2 kHz) [43]. The HL in this family appears to be more stable than average; the observed progression rates were lower (0.1 to 0.4 dB HL at 0.5–4 kHz) than the recently established DFNA6/14/38 annual threshold deterioration of 0.53 dB HL per year (PTA_0.5–4 kHz_), and not statistically significantly different from no progression [57]. This stability likely contributed to the relatively late onset of deterioration of speech perception within this family. However, the limited number of subjects should be considered when interpreting these findings.

Vestibular dysfunction has not been extensively studied in DFNA6/14/38 patients. Previous reports did not demonstrate impaired vestibular function; however, in most of these studies, a complete vestibular examination, with an assessment of all semicircular canals and otolith organs, was not performed. Consequently, these studies cannot confirm the absence of vestibular pathology in DFNA6/14/38. There is one report of vestibular symptoms, although these were self-reported, and dysfunction was not confirmed by electrophysiological vestibular examination [58]. The abnormalities we found on electrophysiological examination mainly involved the otoliths. Otolith function has only been studied twice before in DFNA6/14/38 patients. C-VEMP tests were performed in one study in which no aberrations were observed [59]. The other study describes no abnormalities in utricular and saccular function without specifying their methods [50].

Since the present findings show remarkable aberrant VEMP results in DFNA6/14/38, future vestibular examinations should include testing of all end organs, including the otoliths, using o- and c-VEMP testing according to international standards. The value of complete vestibular examination in phenotype description applies not only to DFNA6/14/38 but also to other forms of non-syndromic and syndromic hereditary hearing loss (HL). In determining whether the otolith abnormalities obtained from the VEMP data in our study are typically *WFS1*-associated or just incidental findings, more extensive vestibular data of affected and unaffected family members are needed, which was beyond the scope of this study.

*WFS1* encodes the Wolframin protein, an 890 or 902-amino acid (Ensembl release 107 [60] transmembrane glycoprotein primarily localized in the endoplasmic reticulum (ER) [10,61]. The function of Wolframin is not yet fully understood. The protein was hypothesized and reported to be involved in Ca^2+^ homeostasis of the cytosol, the ER and mitochondria, in ER stress response, and in mitochondrial health [62,63,64,65,66,67,68,69,70,71,72]. More recently, it was also demonstrated that Wolframin is a vesicular cargo receptor and, as such, involved in the transport of soluble secretory proteins (in pancreatic β cells), including proinsulin, to the Golgi complex for further processing. A defect in vesicular transport subsequently results in ER stress [73]. Wolframin was also shown to function in the Ca^2+^ transfer between the ER and mitochondria. To do so, it complexes with neuronal calcium sensor 1 (NCS1) and inositol 1,4,5-trisphosphate receptor (IP3R). The abundance of NSC1 in neurons suggests that dysfunction of Ca^2+^ transfer contributes to the pathogenesis of *WFS1*-associated disease [74]. Wolframin is ubiquitously expressed in the mouse inner ear, including in various cochlear cells (*i.a.* inner and outer hair cells, IHC and OHC) and vestibular hair cells [75]. However, the function of Wolframin in the inner ear and the molecular pathogenesis of *WFS1*-associated HL still has not been elucidated. The presence of Wolframin in both IHC and OHC in the mouse cochlea may provide cues for rehabilitation. Amplification may be beneficial in an OHC problem while less or no benefit is expected in IHC problems. Several subjects, including the two youngest (aged 5 and 11), use hearing aids and experience real-time benefits from amplification. This may potentially indicate an OHC rather than an IHC problem in this DFNA6/14/38 family.

Interestingly, in *WFS1*-associated HL, hearing can be impaired in the high or in the low frequencies [12]. Wolframin expression in the basal and apical turns of the cochlea is not different and therefore such a difference cannot explain the difference in frequencies in which hearing is affected between Wolfram syndrome and DFNA6/14/38 [75]. Since wolframin is involved in Ca^2+^ homeostasis, a potential role of endolymphatic hydrops in the disease mechanism of DFNA6/14/38 could be hypothesized. A disturbance in endolymphatic ion homeostasis may lead to endolymphatic hydrops due to altered osmotic pressure [76]. Such a pathophysiological mechanism is associated with Ménière’s disease [76], in which low- to mid-frequency SNHL may occur in attacks [77]. It is important to note that in Ménière’s disease, unlike in DFNA6/14/38, HL occurs in attacks; hypothetically, this could be attributed to chronic endolymphatic hydrops in DFNA6/14/38 as opposed to intermittent hydrops in Ménière’s disease. Therefore, MR imaging of the cochlea was performed in the proband, which did not reveal cochlear or vestibular hydrops.

## 5. Conclusions

In conclusion, our observations show that the identified novel *WFS1* variant (c.2512C>T p.(Pro838Ser)) is causal to DFNA6/14/38 with a stable HL of 50 to 60 dB HL in the lower frequencies and individually variable but less prominent HL in the higher frequencies. We suspect a congenital onset and suggest neonatal screening of children in DFNA6/14/38 families with frequency-specific ABR or ASSR testing rather than OAE testing to adequately detect the possible presence of DFNA6/14/38. In addition to HL, our findings indicate mild subjective vestibular dysfunction and absence of otolith functionality in several subjects. Given the limited number of subjects, it is uncertain whether this is typically *WFS1*-related. To further define the vestibular phenotype of not only DFNA6/14/38 but also other forms of hereditary HL, reporting of complete vestibular testing is needed. We suggest including these tests, at least in several subjects, in future genotype-phenotype correlation studies. The findings of our study add to the genotypic and phenotypic spectrum of DFNA6/14/38 and aid in proper patient counseling of individuals diagnosed with this variant in the future.

## Figures and Tables

**Figure 1 genes-14-00457-f001:**
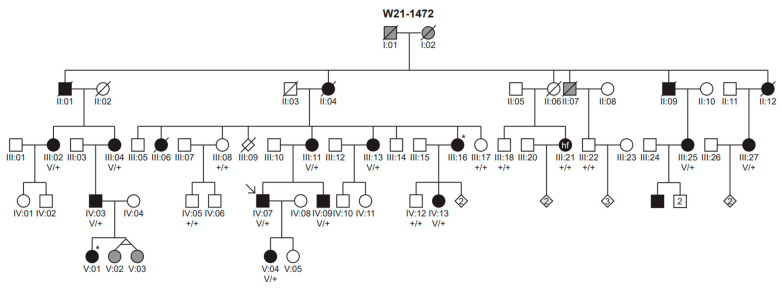
Pedigree of family W21-1472 and co-segregation of the *WFS1* variant with HL. The clinical status of the individuals is indicated by the filling of the symbols: black indicates that the subject has hearing loss, white indicates that the subject has no hearing loss and grey filling refers to an unknown status of hearing. The clinical status is based on audiograms in all participating subjects (with variant status V/+ or +/+) and in III:16 and V:01 (marked with an asterisk), who did not participate but provided an audiogram and who did participate but was not genetically tested, respectively. The clinical status of all other individuals is based on hetero-anamneses. Subject III:21 has a distinctive high frequency (hf) hearing loss phenotype. V, *WFS1* variant (c.2512C>T p.(Pro838Ser)); +, wildtype; square, male; circle, female; slash through symbol, deceased; arrow, index case.

**Figure 2 genes-14-00457-f002:**
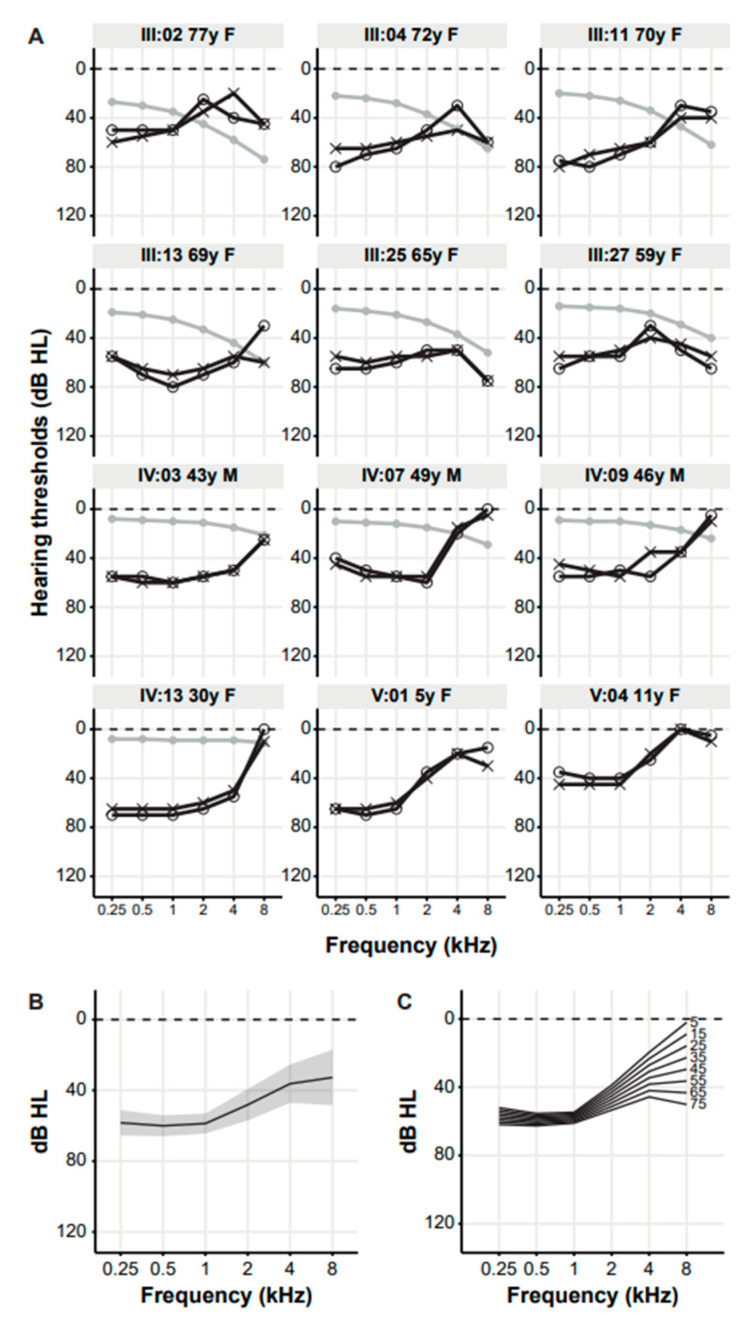
Audiologic features of DFNA6/14/38 subjects from family W21-1472. (**A**) The pure tone air conduction thresholds in dB HL of 0.25 to 8 kHz of all subjects identified with the *WFS1* (c.2512C>T p.(Pro838Ser)) variant (III:02, III:04, III:11, III:13, III:25, III:27, IV:03, IV:07, IV:09, IV:13 and V:04) and subject V:01 who was not genetically tested. Black lines with circles represent the right ear, black lines with crosses represent the left ear, grey lines and dots represent the age- and gender-specific 95th percentile. (**B**) Mean audiogram with a 95% confidence interval. (**C**) Age-related typical audiograms (ARTA), derived from cross-sectional linear regression analysis of the most recent audiograms of DFNA6/14/38 subjects. Each line represents a ten-year age span. dB HL, decibel hearing level; f, female; kHz, kilo hertz; m, male; y, years.

**Figure 3 genes-14-00457-f003:**
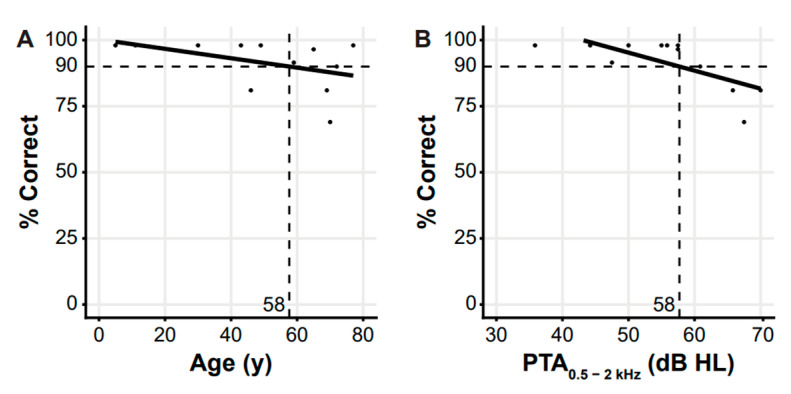
Performance–age (**a**) and performance–impairment (**b**) plots. Cross-sectional analyses are shown in a performance–age plot of means (of both ears) of percentage correct phoneme recognition scores relative to age in years (**a**) and a performance–impairment plot of the same scores relative to pure tone averages (PTA) of 0.5 to 2 kHz (PTA_0.5–2 kHz_) in dB HL (**b**). Panel (**A**) shows an onset age of deterioration of speech perception of 58 years. Panel (**B**) shows an onset level of deterioration of speech perception of 58 dB HL (PTA_0.5–2 kHz_). Continuous lines are linear regression lines; dotted lines relate to 90% correct scores. dB HL, decibel hearing level; PTA, pure tone averages; y, years.

**Table 1 genes-14-00457-t001:** Results of vestibular testing of DFNA6/14/38 subjects.

Subject	III:04	III:11	IV:07	IV:09
Age (y)	72	70	49	46
Vestibular complaints (DHI score)	Yes (NC)	Yes (34)	No (0)	No (0)
Caloric test ^a^				
SPV ^b^ (R/L)	44/36	26/32	18/14	45/14
Rotating chair				
SCV ^b^ (R/L)	38/61	52/NT	31/29	41/27
Tau ^b^ (R/L)	25/17	25/NT	47/55	21/29
GA ^b^ (R/L)	929/1015	1271/NT	1421/1590	857/641
vHIT (mean gain)				
Anterior (R/L)	N/N	N/N	N/N	N/N
Lateral (R/L)	N/N	N/N	N/N	N/N
Posterior (R/L)	N/N	N/N	N/N	N/N
VEMP				
Ocular (R/L)	N/A	A/A	N/N	A/A
Cervical (R/L)	A/N	A/N	N/N	N/N

Abnormal results are highlighted in grey. ^a^ Caloric irrigation test is performed with warm water. ^b^ Normative values in our institute: SPV 10–52°/s, SCV 30–65°/s, Tau 11–26 sec, GA 485–1135°. A, abnormal; CCW, counter clock-wise; CW, clockwise; GA, Gesamtamplitude; L, left; N, normal; NC, not completed; NT, not tested; R, right; SCV, slow component velocity; SPV, slow phase velocity; Tau, time constant; y, years.

## Data Availability

All data relevant to the study are included in the article or uploaded as Appendix A. Data not included in the manuscript or supplemental data file are available upon reasonable request (ronald.pennings@radboudumc.nl).

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
