# Peer review of "Genotype and Phenotype Analyses of a Novel WFS1 Variant (c.2512C>T p.(Pro838Ser)) Associated with DFNA6/14/38"

_genes, 2023, doi:10.3390/genes14020457_

Round 1

Reviewer 1 Report

This is a very nice paper describing the phenotype of a novel WFS1 variant with extensive work-up of a large family  an autosomal dominant form of hearing loss.  That affected children are not picked up by newborn screening is important information as the earlier any accommodation can be implemented, the better the outcome, even if the hearing loss is mild.

There are only a few minor edits needed:

P1., line 45: change "sex-linked" to "X-linked"

P2, line 71: change "This is de last" to "This is the last"

P3, line 114: the sentence that begins "Segregation analysis..." should be changed.  Segregation analysis is a specific computational analysis.  Better phrasing would be "Testing for the variant in 18 relatives was performed by PCR...."

Author Response

This is a very nice paper describing the phenotype of a novel WFS1 variant with extensive work-up of a large family  an autosomal dominant form of hearing loss. That affected children are not picked up by newborn screening is important information as the earlier any accommodation can be implemented, the better the outcome, even if the hearing loss is mild.

We would like to thank the reviewer for the time and suggestions.

There are only a few minor edits needed:

P1., line 45: change "sex-linked" to "X-linked"

We have changed “sex-linked” to “X- or Y-linked”.

P2, line 71: change "This is de last" to "This is the last"

We have changed “de” to “the”.

P3, line 114: the sentence that begins "Segregation analysis..." should be changed.  Segregation analysis is a specific computational analysis.  Better phrasing would be "Testing for the variant in 18 relatives was performed by PCR...."

We have changed the phrasing of “segregation analysis” in the abstract (line 19-20), material and methods (line 115-117) and results (line 196-198).

Reviewer 2 Report

This is a well-written article on a congenital hearing loss.

Introduction - it would be helpful to know the incidence of WFS.

The authors note a variability in hearing loss presentation - predominantly low frequency, but can be high or low (lines 400-412). Are there other genomic anomalies seen in this syndrome besides WFS1 genes? Has GWAS been done on any of these folks, or whole genome sequencing to look for intronic or intragenic variants, or any other genes involved?

All in all, well done.

Author Response

This is a well-written article on a congenital hearing loss.

We would like to thank the reviewer for the time and suggestions.

Introduction - it would be helpful to know the incidence of WFS.

We have added the prevalence of WFS to the introduction (line 60).

The authors note a variability in hearing loss presentation - predominantly low frequency, but can be high or low (lines 400-412). Are there other genomic anomalies seen in this syndrome besides WFS1 genes? Has GWAS been done on any of these folks, or whole genome sequencing to look for intronic or intragenic variants, or any other genes involved?

Our article is about a family with DFNA6/14/38, which typically involves hearing loss in the low frequencies. In contrast, in Wolfram syndrome, the high frequencies are affected. These are two different conditions with different inheritance patterns.

The reviewer's comment is not entirely clear to us. We assume that "this syndrome" means Wolfram syndrome and not DFNA6/14/38. For both conditions, enough research has been done and enough cases are known to state that reported or published missense or truncating variants are pathogenic WFS1 variant(s). Therefore, it is highly unlikely that many of the reported cases have causative intronic variants that are not detected in exome sequencing.

GWAS is useful only in large patient cohorts (thousands of subjects) to identify loci with risk factors for disease (with a complex inheritance pattern in combination with environmental factors), which is not feasible for rare conditions such as Wolfram syndrome and DFNA6/14/38.

All in all, well done.